# Does Diglossia Impact Brain Structure? Data from Swiss German Early Diglossic Speakers

**DOI:** 10.3390/brainsci14040304

**Published:** 2024-03-23

**Authors:** Lea Berger, Michael Mouthon, Lea B. Jost, Sandra Schwab, Selma Aybek, Jean-Marie Annoni

**Affiliations:** 1Department of Neuroscience and Movement Science, Faculty of Science and Medicine, University of Fribourg, 1700 Fribourg, Switzerland; lea.berger2@unifr.ch (L.B.); michael.mouthon@unifr.ch (M.M.); lea.jost@unifr.ch (L.B.J.); selma.aybek@unifr.ch (S.A.); 2Lucern Regional Hospital Sursee, 6210 Sursee, Switzerland; 3Department of French, Faculty of Art, University of Bern, 3012 Bern, Switzerland; sandra.schwab@unibe.ch; 4Functional Neurological Disorder (FND) Research Group, Department of Clinical Neuroscience, Faculty of Science and Medicine, University of Bern, 3012 Bern, Switzerland

**Keywords:** language, brain, MRI, grey matter, diglossia, bilingualism

## Abstract

(1) Background: Bilingualism has been reported to shape the brain by inducing cortical changes in cortical and subcortical language and executive networks. Similar yet different to bilingualism, diglossia is common in Switzerland, where the German-speaking population switches between an everyday spoken Swiss German (CH-GER) dialect and the standard German (stGER) used for reading and writing. However, no data are available for diglossia, defined as the use of different varieties or dialects of the same language, regarding brain structure. The aim of our study is to investigate if the presence of this type of diglossia has an impact on the brain structure, similar to the effects seen in bilingualism. (2) Methods: T1-weighted anatomical MRI scans of participants were used to compare the grey matter density and grey matter volume of 22 early diglossic CH-GER-speaking and 20 non-diglossic French-speaking right-handed university students, matched for age, linguistics and academic background. The images were processed with Statistical Parametric Mapping SPM12 and analyzed via voxel- and surface-based morphometry. (3) Results: A Bayesian ANCOVA on the whole brain revealed no differences between the groups. Also, for the five regions of interest (i.e., planum temporale, caudate nucleus, ACC, DLPFC and left interior parietal lobule), no differences in the cortical volume or thickness were found using the same statistical approach. (4) Conclusion: The results of this study may suggest that early diglossia does not shape the brain structure in the same manner as bilingualism.

## 1. Introduction

Bilingualism, defined as the coexistence of two or more languages within one person [1] and their use on a daily basis, is a widespread phenomenon which is linked to more than 60% of the world’s population [2]. Simultaneous bilinguals learn two languages from birth, while sequential bilinguals learn a native language (L1) and a second, non-dominant and weaker language (L2) later in life [3]. Communicating in a bilingual context requires control of which language must be used and which must not. According to the inhibition control (IC) model of Green, 1998 [4], a bilingual must inhibit the non-target language in order to communicate in the other one. This mechanism also implies the ability to switch between languages and activate the language needed [5]. The process of switching seems to share features with other cognitive control systems [6]. Another model, bilingual interactive activation (BIA), has been developed in language comprehension, and proposes that words of both languages are stored in a single, shared lexicon, and that similar words in both languages, the relevant and the irrelevant one, are activated when a word is presented [7,8]. A growing literature suggests that bilingualism can be considered as a spectrum with multiple components, such as the age of second language acquisition (L2 AoA), L2 exposure and proficiency [9,10,11]. The proficiency also influences the amount of inhibitory control that is activated, as explained in the IC model [4]. With an increased proficiency in an L2, there is a shift to a more automatic control system and easier language processing [12]. However, expressing oneself in different idioms does not necessarily imply bilingualism. 

Another linguistic phenomenon is diglossia, the use of different varieties or dialects of the same language. Typically, one variety is a literary, more formal and written language, while the other language is a spoken form for everyday informal communication [13]. Prime examples of diglossia are the different forms of Arabic, modern Greek and German/Swiss German [14].

Although similar neural circuits seem to be shared in first and second language processing, their implication tends to be modulated by linguistic context. A recent metanalysis [15] showed a larger recruitment of dorsal-stream regions during phonological processing [15] and a larger activation of executive networks during semantic processing in bilingualism when L2 was compared to L1 [16], as well as an increased implication of neural networks in semantic tasks for L1 (i.e., left lateral and ventral temporal cortex parietal and inferior frontal gyrus) [16]. Moreover, communicating in a bilingual context implicates the “language control” system—particularly including the left anterior cingulum, dorsolateral prefrontal cortex and caudate [17]—as part of the general cognitive control network [6,18,19,20,21,22]. Abutalebi (2008) [23] and Branzi et al. (2016) [21] have revealed a functional involvement of the left inferior frontal gyrus (IFG), caudate nucleus (CN), the anterior cingulate cortex (ACC), dorsolateral prefrontal cortex (DLPFC) and the inferior parietal lobule (IPL). These regions were also shown to be important in the diglossic setting of the Arabic language with Hebrew [24]. Furthermore, control systems of speech production and articulation are also important, such as the planum temporale [25,26].

Acquiring a second language has a plastic influence on focal cortical densities [27]. Differences in the cortical volume between bilinguals and monolinguals have been consistently reported through structural MRI analyses of the left inferior frontal gyrus (IFG), middle and superior frontal gyrus and the anterior cingulate cortex (ACC), as well as the parietal cortex [27,28,29,30]. An increase in grey matter density has been reported at the level of the ACC [31], the DLPFC [32] and left caudate nucleus [33]. In bilingual participants, Elmer et al. [34] observed an initial increase in grey matter density followed by a reduction as participants become more proficient. Also, L2 immersion leads to expansion of subcortical structures such as the left striatum and the thalamus [9,35,36]. 

Whether diglossia is a form of bilingualism [24,37,38] is a matter of debate. Findings that young preliterate children with a mother tongue of spoken Arabic (SA) are not able to understand a simple story in Arabic literature (LA) have led to the suggestion that diglossia can be considered as a form of bilingualism. Moreover, the finding that diglossic and classical bilingual children show similar metalinguistic abilities [39] also suggests that diglossia can be considered as a form of bilingualism. 

Swiss German is a dialect spoken by the German-speaking community in Switzerland. Some experts consider Swiss German (CH-GER) as a “mixed language” together with standard German [40] because these two languages are closely related and share part of their vocabulary [41]. Others postulate that Swiss German/standard German is a type of diglossia, as is the case in the Arabic language [14,42]. Swiss German speakers use two varieties of the same language, standard German (stGER) and CH-GER, in different socio-linguistic situations. For example, Swiss German speakers use CH-GER in their family and within group social life, but all the teaching, professional and official exchanges, as well as reading and writing, are in stGER. CH-GER is considered the mother tongue of children in the German speaking part of Switzerland [14]. Understanding stGER necessitates that CH-GER children learn to recognize and selectively respond to subtle differences in these two forms of German. Bühler and colleagues [43] classified such similarities and differences into three groups: (i) words that are exactly the same in CH-GER and GER such as “Baum” (tree) and “Fisch” (fish); (ii) words with a different pronunciation as in “Tiger” (stGER: [ˈtʰiːɡɐ] CH-GER: [ˈtiɡ̊əɾ]) (tiger) and “Adler” (stGER: [ˈaːdlɐ], CH-GER [ˈɒd̥ləɾ]) (eagle) where the vowels are prolonged in CH-GER; and (iii) words that are not at all the same, like “Rahm” (CH-GER)/“Sahne” (GER) (whipped cream), “Rüebli”/“Karotte” (carrot) and “Chappä”/“Mütze” (cap). Moreover, there are differences as well as similarities at the phonologic, morpho-syntactic and grammatic levels [14,44,45]. In school, Swiss German children learn how to talk in stGER as well as to read and write in stGER, because there is no official grammar in Swiss German [43]. It has been shown that the first exposure to stGER is not during school; it starts much earlier in childhood with television and other media [46]. Despite some shift towards a phonetic written form of CH-GER in younger people due to social media utilization [47], there is a consensual opinion that CH-GER/stGER can be considered as a form of diglossia [48].

Some behavioral and functional studies on diglossia raise the question of whether communicating in formal or informal contexts may modulate linguistic processes and the amount of activation of language and control networks. According to the IC model of Green (1998) [4], the high proficiency and the proximity between the standard and the familiar language in diglossia can influence the nature of the language control system. Some studies have tried to answer this question by analyzing behavioral responses and brain activation through the functional MRI paradigm. One example is the diglossic situation of spoken (SA) and literary (LA) Arabic, as there is a faster process in LA while reading words [37] and a faster process in SA in an auditory or oral modality [49]. In their behavioral and neuroimaging study, Abou-Ghazaleh et al. [24] asked the participants to name each image either in SA or LA while brain activity was recorded through fMRI. As mentioned earlier, neuroimaging analysis showed no difference between SA and LA [24]. In another study, which was more oriented towards language selection, fMRI showed two distinct patterns of differences in activation between diglossic and bilingual contexts [13]. This last study suggests that there may be a specific language control system associated with diglossia. Studies on dialect switching in diglossic speakers showed both behavioral slowing and electrophysiological activation during switching between dialects [50] in regions such as the left DLPFC, ACC IPL and basal ganglia [51,52]. Yi et al. [50] suggest the implication of the language control system when switching between dialects. 

The impact of diglossia—and particularly early diglossia (i.e., learned in the first years of life)—on brain structure is still unknown. To our knowledge, there are, to date, no studies that focus on eventual structural brain changes induced by diglossia, in contrast to bilingualism. Such studies are legitimate, given the studies on the Arabic language conducted by Abou-Ghazaleh et al. [13,24]. In this study, we focused on the brain structure and whether the changes observed in the functional analysis can have an impact on a structural level in the brain. Therefore, the present study aims to examine brain structures of an equivalent diglossic and non-diglossic group of Swiss university students. To this end, we took advantage of the local linguistic context. The majority of university students at Fribourg University are Swiss students, approximately half with a Swiss German native background and half with French native background. Most of the university students are actually multilingual, since they receive courses in French, German and English at school. However, the Swiss students with a CH-GER linguistic background are diglossic and they are exposed to stGER at a very early stage in life. As there is no French dialect and the Swiss students with a French background (CH-FR) do not speak CH-GER at an early stage in life and study stGER only later, they are non-diglossic. Their immersion in CH-GER is negligible during studies in Fribourg and begins only later in life, such as in professional life. In the area of Fribourg, the language spoken outside university is predominantly French. In CH-FR, there are no differences in spoken and literary French. So, between these bilingual/multilingual groups, the major difference is the early diglossia which is present only in the CH-GER group. In the present MRI study, we looked at differences in the cortical thickness and grey matter volume in the whole brain as well as five regions of interest (ROIs) defined based on previous studies [5,21,23]. Based on the findings by Abou-Ghazaleh et al. (2018, 2019) [13,24] in their functional neuroimaging and behavioral studies, we expect to find structural changes in language control systems in diglossia but not necessarily differences in language-related brain networks in the CH-GER and CH-FR groups. The question addressed here is whether early diglossia, in a bilingual context (French and German), leads to modification in the brain structure.

## 2. Materials and Methods

### 2.1. Participants

Structural MRI data of 44 CH-GER and CH-FR participants, acquired at the University of Fribourg from two separate studies, were merged for this study: 28 participants from dataset 1 [53] and 16 from dataset 2 [54]. In the dataset of this study, two groups were defined: the first group included the native CH-GER-speaking participants and in the second group, native CH-FR-speaking participants were included. The diglossic CH-GER group had learned to speak the CH-GER dialect from birth and was exposed to (and also started to learn stGER) before the age of 6 and before entering kindergarten [46]. The non-diglossic CH-FR group was only exposed to standard French at a young age. Both groups learned their formal second language after the age of 6. These L2 languages were mostly English and French for the CH-GER group and standard German for the CH-FR group (for details about the spoken languages see Table 1). Exclusion factors were early bilingualism, early stGER exposure for CH-GER participants (such as being a child of standard German-speaking parents) and a history of neurological or psychiatric disorders. Out of these 44 participants, 2 were removed due to movement artifacts in the MRI scans. Participation was voluntary and, if matching the criteria, they were selected for the two studies described above. Their personal and schooling background was controlled through an internally developed questionnaire on their education, age, sex and health. The final number of participants was 42 (29 females), who were all healthy, late multilinguals aged between 19 and 45 (mean age ± standard deviation [SD] = 23.81 ± 5.38). Concerning the power analysis, we first referred to one of the seminal papers on brain structural differences between monolingual and bilingual speakers, i.e., Mechelli et al.’s analysis [27]. In their research on testing for differences in the density of grey and white matter between bilinguals and monolinguals, they recruited 25 monolinguals who had had little or no exposure to a second language and 25 early bilinguals. They found specific differences in one predicted area (increased left inferior parietal cortical density in bilinguals). We thus performed an analysis of statistical power with the G Power paradigm [55]. To detect differences between two groups, it was concluded that, in order to achieve a power of 0.80 (one tail, effect size d = 0.80, α = 0.05, details of the power analysis in Appendix A), a sample size of 21 participants per group should be used. So, we concluded that our sample was adequate compared to the bilingual literature. To test the differences in the density of grey and white matter in relation to proficiency and age of acquisition, Mechelli et al. [27] compared 25 monolingual and 25 bilinguals. So, we concluded that this sample was adequate for our specific study.

The study contained two groups: the first group with 22 participants with CH-GER as their first language (17 females/5 males, mean age ± SD = 23.86 ± 6.98) and the second group with 20 participants with CH-FR as their first language (12 females/8 males, mean age ± SD = 23.75 ± 2.95). In addition to their L1 and to the dialect for CH German, they all learned a second language (L2) (French or German) at the age of 9.8 ± 1.7 years on average, and continuously learned this as an L2 in addition to acquiring at least one other language during their life (average number of spoken languages: 3.4 ± 0.7). All of the participants had at least 12 years of education, with an average of 16.1 ± 2.5 years. The participants were all right-handed according to the Edinburgh handedness inventory [56] and did not report any history of neurological or psychiatric disorders. The study protocol was approved by the local ethical committee (Fribourg 020/530).

To evaluate language abilities, the usage of L1 and L2 and the level of self-reported proficiency in L1 and L2, the participants had to fill out the Language Experience and Proficiency Questionnaire (LEAP-Q) [57]. This questionnaire evaluates language profiles of multilinguals based on language competence (proficiency, dominance and preference), the age of language acquisition, the method of language acquisition and past and present language immersion. In the questionnaire, some of the participants did not fill out the form correctly and some data were missing. To be able to still perform the analysis, the missing data were replaced using a value that was calculated using the mean values of all of the existing data of the other participants. This value then was filled into the missing gaps. This method is called a correction by imputation [58]. To signal which results have been corrected using imputation, they are marked with ^1^ and ^2^ in Table 2. The descriptive statistics of the population are reported in the results section.

### 2.2. Structural MRI Data Acquisition

For the MRI images, a 3T MRI scanner (Discovery MR750; GE Healthcare, Waukesha, WI) was used with a 32-channel standard head coil at the Hospital of Fribourg HFR. To reduce head movements, the head of the participants was placed into fixation foam during the scan. A high-resolution, T1-weighted anatomical scan was performed in the coronal plane from anterior to posterior within 270 slices of a voxel size of 0.86 × 0.86 × 1 mm (FSPGR BRAVO sequence, acquisition parameters: matrix size: 256 × 256, TR = 7.3 ms, TE = 2.8 ms, Flip angle = 9°, Prep Time = 900 ms, parallel imaging acceleration factor = 1.5, intensity correction: PURE). More detailed acquisition parameters can be found in the Appendix A.

### 2.3. MRI Data Preprocessing 

The grey matter was characterized by volume via voxel-based morphometry (VBM), and by cortical thickness via surface-based morphometry (SBM). Both analyses share common preprocessing steps using automated procedures in the Computational Anatomy Toolbox (CAT12.8.2) (the Structural Brain Mapping group, Jena University Hospital, Jena, Germany) implemented in SPM12 (Statistical Parametric Mapping, Institute of Neurology, London, UK) running on MATLAB R2020b (MathWorks, Natick, MA, USA). First, the anatomical image (T1 weighted image) was visually inspected, and its origin was set on the anterior commissure. After bias correction, it was segmented into grey matter (GM), white matter (WM) and cerebrospinal fluid (CSF). The empirical image was assessed the and preprocessing quality based on the resolution, noise and bias was determined using the CAT12 toolbox. In the present dataset, the weighted average index had a 76–84% range (mean ± SD = 80.9 ± 1.46%), indicating medium- to good-quality data. There was no difference in image quality between the two groups (CH-DE: 80.9 ± 1.67%, FR: 80.9 ± 1.23%, two-samples *t*-test *p*-value = 0.9). In the case of VBM processing, GM images were warped and modulated to fit into CAT12′s template brain in the Montreal Neurological Institute standard space (MNI-space). After this step, additional visual and sample homogeneity checks were performed. Finally, images were smoothed with an isotropic 8 mm full width at half-maximum (FWHM) Gaussian kernel.

In the case of SBM processing, tissue segmentation was peformed in the CAT12 toolbox to estimate the distance between the inner surface (WM/GM interface) and the outer surface (GM/CSF), the distance corresponding to the cortical thickness. The local maxima of this distance was then projected onto other neighboring GM to create a cortical thickness map [59]. This approach allows for handling partial volume information, sulcal blurring, and sulcal asymmetries without explicit sulcus reconstruction. For inter-subject comparisons, cortical thickness maps were resampled into a common coordinate system and smoothed using a 15 mm Gaussian heat kernel [60,61]. 

### 2.4. Analyses

We conducted two different structural analyses: voxel-based morphometry (VBM) and surface-based morphometry (SBM). They were both conducted on the whole brain and on a group of regions of interest according to Green and Abutalebi’s theories. Moreover, a voxel-wise or a mean-wise analysis could lead to different results. This is why we also decided to conduct a voxel-wise analysis restricted to our region of interest in addition to the whole-brain analysis.

In the VBM analysis, preprocessed images were studied in a general linear model (GLM) using the random effect approach (RFX). This model permits a comparison of the two groups with a two-sampled *t*-test assuming unequal variances. The model included age and the total intracranial volume (TIV) as confounding factors. As the TIV and age are known to have an impact on brain volume [62], they were considered as an effect of no interest to ease interpretation of the results. Gender was not included because it is correlated with the TIV (Spearman correlation ρ = −0.4, *p* < 0.001). This analysis was first conducted at the whole-brain level with an absolute mask threshold of 0.15 and then only in the regions of interest (ROIs) (see the Section 3).

For the SBM analysis, a similar procedure was used on the preprocessed data but with age and gender as effects of non-interest. No implicit mask was used in the whole-brain analysis and explicit ROIs were studied in a second step (see below). 

For further analyses, regions of interest (ROIs) were defined based on the bilingual literature (see Section 1). Therefore, the following ROIs were outlined: the left inferior frontal gyrus (IFG) and caudate nucleus (CN) [5,21,23], as well as the anterior cingulate cortex (ACC) [63], dorsolateral prefrontal cortex (DLPFC) [32], the inferior parietal lobule (IPL) [5,21,23] and planum temporale [25,26]. For the VBM analysis, the ROIs used were the bilateral ACC, bilateral caudate and left DLFPC (which included the left inferior frontal per opercularis-trangularis, middle frontal, superior frontal), defined based on the neuromorphometrics probabilistic atlas provided by SPM12 (MRI scans originating from the OASIS project (http://www.oasis-brains.org/, accessed on 17 March 2023) and the labeled data were provided by Neuromorphometrics, Inc. (Winthrop, MA 02152-1083 USA), (http://www.neuromorphometrics.com, accessed on 17 March 2023 under an academic subscription). The left IPL was defined based on the Julich Brain Atlas (Human Brain Project: https://www.humanbrainproject.eu/en/follow-hbp/news/2021/10/13/complete-data-package-julich-brain-atlas-released/, accessed on 17 March 2023), and finally, the left planum temporale was defined based on the SPM Anatomy toolbox version 2.2b [64,65,66]. For the cortical thickness analysis, the ROIs covered all the same cortical regions. The caudate was not considered since it is subcortical. In the first step, a voxel-wise analysis of the ROIs was performed using VBM and SBM to compare the two groups. In the second step, the pondered means (mean-wise analysis) only for the grey matter volume (first Eigenvariate) for each sub-region were extracted. As these mean data did not lead to any conclusion with a traditional parametric statistical analysis (ANCOVA), a Bayesian ANCOVA with these values was conducted to better characterize the difference between groups. The Bayesian ANCOVA was performed instead of a direct comparison between the means of the two groups in order to remove the effects of confounding factors (age and TIV) [67]. This was performed to outline the chance of similarity of the two comparing groups; therefore, we kept the 0-hypothesis (H0) and rejected the alternative (H1) one. With the BF01, there was a much higher chance that two comparable group factors were the same; therefore, we kept H0 and rejected H1 [68]. 

For all the voxel-wise analyses (whole brain and ROIs), a statistical threshold of p_FWE_< 0.05 family-wise error, corrected for multiple comparisons at a peak level, with the Threshold-Free Cluster Enhancement Estimator (TFCE toolbox under SPM12), was used to increase the sensitivity of the results [64]. Anatomical locations were checked using the neuromorphometrics probabilistic atlas provided by SPM12. All the coordinates derived from these analyses are given in the MNI space. For the mean-wise analysis, the Bayesian ANCOVA was performed using the jsq module 1.0.2 (The JASP Team, Damian Dropmann, Ravi Selker, and Jonathon Love) in Jamovi 1.6.7 (The Jamovi Project, URL access in https://www.jamovi.org, accessed on 16 March 2023. Analysis files can be found in the repository mentioned in Appendix A.

## 3. Results

### 3.1. Descriptive Statistics of the Population

The two groups, namely the CH-GER and the CH-FR group, did not differ between different categories in terms of gender (chi-squared 0.94, *p* value 0.23), age, years of education, L2 proficiency and daily exposure to L2. Overall, they also did not differ with regard to family, friends, reading and TV. 

Differences between the two groups were found in the daily exposure to L1 (higher exposure to L1 in the CH-FR group), the age of L2 acquisition (earlier for CH-FR) and in the daily exposure to L2 during self-study (higher in the CH-GER group). For further details, see Table 2. 

The age of L2 acquisition was earlier and the daily exposure to L2 during self-study was higher in the CH-GER group. For further details, see Table 2. 

### 3.2. Structural MRI Data 

*Whole-brain analysis*: In the *t*-test to compare the grey matter volume (VBM) and cortical thickness (SBM), no statistically significant anatomic differences between the CH-GER and CH-FR groups were observed. 

*Comparisons between Regions of Interest*. The ROIs are illustrated in Figure 1. Results (shown in Table 3) of the regions of interest (ROI) voxel-wise analysis did not show any statistical differences either in grey matter (VBM) or cortical thickness (SBM). Also, an analysis of the sub-ROI grey matter volume was separately performed mean-wise with a Bayesian statistic. 

### 3.3. Sub-Analysis

All five ROIs showed an identical prior probability P(M) for the alternative hypothesis compared to the null hypothesis. The error percentages were all below 0.02%, with a mean of 0.014162%. The results are therefore acceptable due to an error % of less than 20%.

In this analysis, we looked at the BF01, which is the ratio between the likelihood of the data fitting under the null hypothesis (H0 = no group difference) to the likelihood the data fitting the alternative hypothesis (H1 = difference between CH-FR and CH-GER). In other words, this value reflects if H0 is more likely than H1 regarding this set of data. The left IPL and the left planum temporale both had a Bayesian factor (BF01) close to 1, so no conclusion could be made whether there is a difference or a similarity between the two groups (equal posterior probability of H0 and H1). The three other ROIs, namely the left DLPFC, the bilateral ACC and the bilateral caudate, showed a BF01 larger than 1; therefore, there is a higher chance that the two groups have the same volume (H0) than a different one (H1) [68]. 

## 4. Discussion

The aim of this study was to investigate if diglossic immersion leads to changes in brain structure. We therefore analyzed if there were differences in the brain structure between German-speaking diglossic and French-speaking non-diglossic students living in the same area and studying at the same university. Note that the only difference in language background in our study between both groups was the presence of diglossia in the CH-GER group and not in the CH-FR group. It is worth mentioning that there are other factors that might influence changes in the MRI that have not been compared between the two groups, namely socio-economic status [70], musical experience [71], intelligence [72], process speed, verbal fluency and executive functions [73,74,75]. The analyses showed no difference in the grey matter volume and cortical thickness between the Swiss German diglossic and French-speaking non-diglossic participants. A supplementary analysis of two matched subgroups in gender, and L1 and L2 exposure yielded similar results (Appendix A). We acknowledge that the lack of difference does not necessarily indicate equality of the CH-GER and CH-FR groups. Bayesian analyses indicate a higher chance of a similar volume between groups than a difference. This similarity was found in the whole-brain analysis as well as in the analyses of the ROIs specifically implicated in language control and the language network. Such results suggest that diglossia has no significant plastic influence on the brain structure in a group of already multilingual participants, and, to the best of our knowledge, these results have not been reported yet. They are partly in line with data from functional activation studies, where no differences between dialect and standard language were reported. In particular, Abou-Ghazaleh et al. [13] found similar brain activation between SA and LA in picture naming, even though naming a picture correctly was faster for SA compared to LA. The main results of the present study concerning the left DLPFC, the ACC and the bilateral caudate showed a low, but still prevailing, chance of similarity between the two groups. These findings appear to support an earlier study by Abou-Ghazaleh et al. [24] on fMRI during a picture naming task, which showed no difference in the activation of the ACC, SMA and caudate nucleus between SA and LA. Naming in the official L2 (Hebrew) led to higher activation compared to SA, but naming in Hebrew was performed at the same speed as naming in LA. According to the authors, one reason for these differences could be that both languages act somewhat as an L2 depending on the linguistic context (written, spoken or auditory) [24].

Contrary to these studies on naming tasks, research in the classical bilingual context (i.e., French/German or French/English) has repeatedly shown a higher activation of language networks and cognitive neural systems when participants process words in an L2 context [76,77], as well as brain plastic modifications [27]. In the IC model of Green (1998) [4], it has been predicted that depending on the dominance of the language, inhibition takes place as there is a larger switching cost for the stronger language [78,79,80]. On the other hand, it has been said that when the dominance between L1 and L2 is similar, the switching costs are comparable [81,82]. We suggest that, in diglossia, both CH-GER and stGER can be considered as an L1 depending on the situation; this then would lead to minimal switching costs and thus minimal structural changes in the associated brain areas. Also, diglossia may also apply to the BIA model [7,8], where a similar baseline activation can occur and lexical differences are minimal between both L1s (CH-GER and stGER). Therefore, word activation is determined by the context. The strong link between CH-GER and stGER can also justify why there is little mediation mechanisms in the sense proposed by the revised hierarchical model (RHM) [83]. Also, the conceptual system of the two diglossic lexicons is highly linked, and producing a word in stGER does not need to be mediated by CH-GER, like it would be the case in a less similar L2 [83,84]. The question of intralanguage control mechanisms must be further studied. Similarly, structural studies have shown differences in brain volume between monolinguals and early bilinguals in various executive areas such as the IFG, MFG and ACC as well as in the parietal cortex [27,28,29,30]. As already mentioned, structural changes in grey matter density have been observed [31,32,33]. When comparing bilinguals and multilinguals who are early simultaneous bilinguals and acquired a third language later in school, some authors have shown that knowing three languages leads to a higher grey matter density but not volume in the right supramarginal gyrus [85]. In line with this, looking at the grey matter volume using VBM, studies have shown a volume increase in the cingulate cortex, as well as in the parietal and frontal areas [86]. These changes were associated with the age of acquisition and L2 ability. More in line with our findings, and as shown by other bilingual studies, is that due to high proficiency in an L2, there is a pruning of the grey matter after an initial increase [87,88]. For white matter changes, according to our results, speaking the standard language plus the dialect does not necessarily lead to structural cortical or subcortical differences. Together with the data from the studies of Lövdén et al. (2013) [88] and Wenger et al. (2017) [87], our results suggest that the Swiss German dialect and standard German may not necessarily be perceived as a distinct second language by the brain, but rather as a variety of a single language. Our conclusion is, however, cautious; all participants were multilingual, young, healthy students with a high-quality and lengthy education. It might be possible that the changes in morphometry in diglossia are small and that the changes due to multilingualism, as seen in our participants, outweigh the effects of diglossia.

Changes in bilinguals have also been reported when looking at the white matter, with changes in the microstructure and organization of the fibers in the tracts [89]. In MRI, these changes are measured using fractional anisotropy (FA), which is highly sensitive to changes in the microstructural architecture [90]. With that in mind, a supplementary analysis on white matter changes might help to detect smaller changes in diglossia. 

An alternative mechanism could be examined to explain the absence of significant statistical differences in the cortical volume between diglossic and non-diglossic participants in the present study. First, there are strong linguistic similarities between the CH-GER dialect and standard German (stGER). It has been suggested that the linguistic distance between two languages has an impact on the control system [91]. Kim et al. [92] showed similar brain activations when the languages are very similar, while dissimilar networks have been reported between distant languages. In order to test such a possible explanation, functional studies on direct comparisons of brain activation between speaking or listening in CH-GER and in stGER will have to be performed in the future. The phonological and syntactic similarities between CH-GER and stGER may hinder the plastic changes usually seen in bilingual situations. To identify phonological differences, bilinguals rely on the auditory–articulatory mapping of the planum temporale [93,94,95]. If the differences between two varieties of one language are small, this might not lead to structural changes in the planum temporale, but increasing the language distance could increase activation differences [96]. 

A second alternative explanation is based on the high level of proficiency of the diglossic subjects in CH-GER and stGER. If one is highly proficient in both languages, switching between these languages is an automatized process [22], so that cortical changes are less likely to occur. Claussenius-Kalman et al. [97] have shown a thinner cortex in language control areas in bilinguals with an equal proficiency in L1 and L2, but an initial expansion of cortical regions when learning a new language [87,88]. Some authors suggest that there is cortical pruning [87,88], while others suggest there is cortical thickening [98] as a function of proficiency. This, in turn, complicates the interpretation of the cortical changes in diglossic subjects. 

In the current study, we identified some limitations and weaknesses. Firstly, both groups, the diglossic and non-diglossic groups, consist of multilinguals, with a mean number of spoken languages of 3.4. All Fribourg university students are de facto multilingual, as they are obliged to communicate at least in French, German and English, and this could possibly mask a diglossic effect. On the other hand, we cannot be sure that the French-speaking group was not also exposed to Swiss German, even though we believe that this exposure will be quite limited as, for example, the media is mostly in stGER. To be able to exclude this possible impact, we would have to perform the analysis with a group outside of Switzerland. There were some differences in the groups, such as the age of acquisition (AoA) of an L2, due to some regional differences between schooling programs. For the AoA, the CH-FR group learned an L2 before the CH-GER group. According to some authors, the later one acquires an L2, the thicker the left and the thinner the right IFG cortex becomes [30]. Others have observed that the earlier an L2 is acquired, the thicker the cortical thickness is in the right lateral occipital region, the left superior parietal lobule, the middle temporal gyrus and the left parahippocampal gyrus [99]. Secondly, our groups were different with regard to exposure to an L1, with the CH-FR group more significantly exposed to French, their L1, during the day than the CH-GER group was exposed to German (stGER and CH-GER). This might be due to the predominantly French-speaking city of Fribourg (despite officially bilingual), where the study took place. Exposure can lead to changes in the cortical structures. Stein et al. [100] showed that exchange students in an enriched environment have an increased grey matter density in the left anterior temporal lobe. Based on these findings in different studies and the fact that our two groups were not identical in factors that can lead to structural changes, the findings related to diglossia might be influenced. There might be a lack of power due to a small simple size. After a comparison with work by Munson et al. [101], mentioned before, and a calculation of the Cohen’s d effect size to detect any differences, we require a total group size of 218 to detect a medium effect size. Thus, to sum up, with our sample size (44 participants), we conclude that there are no large differences between our two groups. If we want to detect any small differences between the two groups, a five-times-larger sample size would be needed. We acknowledge that the study of Munson et al. [101] is an analysis based on bilinguals and monolinguals and we have a diglossic dataset. Nevertheless, it is similar to our research. If there was a clear difference between our two groups, we would be able to detect a difference based on the number of participants. The fact that Mechelli [27] could detect differences in bilinguals with a number of participants similar to ours suggests, we believe, that our sample size may be adequate to detect significant differences. Indeed, to detect minimal differences, we would need a much larger participant dataset. Since we found no differences in volume, its impact on structural changes seemed to be minor or undetected. To be able to detect minor changes, additional tools, such as MRI techniques for white matter changes, possibly with a higher sensitivity [90]; multimodal parcellation analyses with an improved statistical sensitivity by minimizing the complexity of the data [102]; or multi-contrast segmentation, will be of particular interest for better tissue characterization (functionally and anatomically) [103] and could possibly detect smaller differences.

Finally, we did not compare executive abilities between both groups. This could have given us supplementary measures of an eventual impact of diglossia on brain functions. Such a cognitive approach would be of high interest in future studies, given the established relationship between language control and executive function.

## 5. Conclusions

In conclusion, we could not demonstrate reshaping of cortical brain structures—usually described in bilingual speakers—in our diglossic population. This raises the question, in line with the results of the other studies, of whether Swiss German/German diglossia may be considered univocally as a classical bilingualism as it induces, in this case, fewer neural adaptations. On the other hand, it is possible that the bilingualism of the participants outweighs the changes based on the diglossia. The close similarity between Swiss German and stGerman may also participate in this absence of demonstrated differences. To obtain a precise answer to this question, further studies, particularly prospective studies with larger groups and comparisons of participants with similar standard L1s, such as diglossic CH-/stGER and non-diglossic stGER speakers, will have to be conducted.

## Figures and Tables

**Figure 1 brainsci-14-00304-f001:**
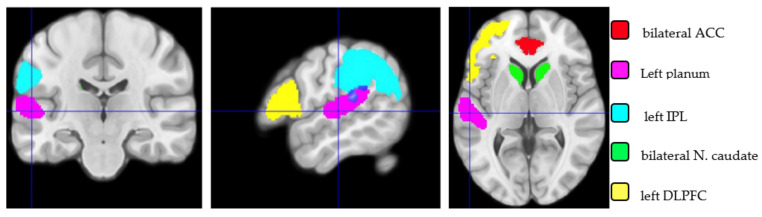
Illustration of the five regions of interest (ROIs): bilateral ACC, left planum temporale, left IPL, bilateral nucleus caudate and left DLPFC. The images are displayed using the neurosciences convention; the left hemisphere is on the left side.

**Table 1 brainsci-14-00304-t001:** Spoken languages of the participants, as mentioned.

	L1 CH-GER					L1 CH-FR					Total Number of Subjects Talking the Language
Language	L1	L2	L3	L4	L5	L1	L2	L3	L4	L5	
CH-German	22										
stGerman	22						17	3			42
French		12	10			20					42
English		10	11				2	17	1		40
Italian		1	1	2			1		1		6
Spanish				2					3	1	8
Latin			1								1
Dutch				1							1
											average number of spoken languages Ø 3.4 ± 0.

**Table 2 brainsci-14-00304-t002:** Group information of the Swiss German (CH-GER, *n* = 22) and French (CH-FR, *n* = 20) first language speaking group. ^1^ imputation of one variable, ^2^ imputation of two variables [58]. Statistics were calculated with dependent *t* tests, except for gender (chi square. * in gender indicates the number of females and males in each group).

Variable	L1 CH-GER		L1 CH-FR		
	Mean	SD	Mean	SD	*p* Value
Age [years]	23.86	6.98	23.75	2.95	0.94
Gender (Female/Male)	5/17 *		8/12 *		0.23
Number of spoken languages	3.41	0.73	3.30	0.57	0.59
Years of education	15.48 ^2^	2.43	16.8	2.42	0.08
Age of L2 acquisition [years]	10.81 ^1^	1.84	8.75	1.33	<0.001
Current L1 exposure [% of the day]	64.64	14.48	74.3	11.50	0.02
Current L2 exposure [% of the day]	17.14	15.87	10.10	8.04	0.08
Subjective language proficiency L2 [0—none, 10—perfect]					
Speaking	5.05	2.01	5.30	1.72	0.66
Reading	5.91	1.60	6.25	1.71	0.51
Understanding	5.95	1.84	6.55	1.67	0.28
L2 Exposure [0–10 scale]					
Friends	2.95	2.92	3.00	1.97	0.95
Family	0.82	1.71	0.45	0.06	0.35
Reading	3.41	2.36	2.65	1.27	0.20
Self-study	1.68	1.96	0.25	0.64	0.003
TV	1.50	1.37	1.00	0.97	0.18

**Table 3 brainsci-14-00304-t003:** Results of VBM value across the ROIs for each group. The second column shows the mean and standard deviation (SD). The third column shows the group factor for the Bayesian ANCOVA with the Bayes factor (BF_01_ = how much more likely the no group difference is than a difference between groups) and the error percentage [69]. Results are independent from the confounding factors (age + TIV) used as nuisance variables in the Bayesian ANCOVA. All values were directly reported from Jamovi 1.6.7 software.

	L1 CH-GER		L1 CH-FR		Bayesian ANCOVA Group Factor	
Regions of Interest	Mean	SD	Mean	SD	BF_01_	Error%
Left IPL	0.588	0.0603	0.617	0.0587	1.19	0.00881
Left planum temporale	0.575	0.103	0.615	0.0782	1.5	0.01255
Left DLPFC	0.609	0.07	0.615	0.0496	3.16	0.01372
Bilateral ACC	0.633	0.0915	0.648	0.0692	2.88	0.01393
Bilateral caudate	0.706	0.0771	0.730	0.0752	2.21	0.01505

## Data Availability

Raw data and materials are accessible at the following public repository, https://www.doi.org/10.5281/zenodo.10814781 (accessed on 22 January 2024) through authorization of the authors, according to Swiss National foundation rules.

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
