# Peer review of "Does Diglossia Impact Brain Structure? Data from Swiss German Early Diglossic Speakers"

_brainsci, 2024, doi:10.3390/brainsci14040304_

Round 1

Reviewer 1 Report

Comments and Suggestions for Authors

The manuscript reports on a study comparing the grey matter volume/cortical thickness between native German speakers (who are fluent in both CH-GER and stGER) to native French speakers. The German speakers were diglossic speakers, that is, have learned standard German language (school) as well every day spoken German language (at home). The French speakers did not have such variation in language use. A VBM analysis was performed and no statistically significant differences were found between groups. In additional cortical thickness was compared – no differences detected. These analysis were followed up with Bayesian approach – no differences detected.

The issue of gender distribution is confusing (page 7) – at one point it is stated there is difference in gender distribution between both groups and at other point it is stated there is not. Please include gender ratio in table 2. The statistical test needs to be corrected – a t-test was used to compare gender rations; in reality a chi-test needs to be performed.

Do the French speakers learn stGER or CH-GER? If they only learn stGER they may still be familiar with CH-GER because they are living in a bilingual environment and thus have exposure to CH-GER a9fn probably acquired knowledge). Did the authors acquire data in regards CH-GER ability (or comprehension ability) in the French speakers? The authors conclude that lack of differences means that diglossic ability does not seem to alter grey matter. But it may be that the French speakers do have some ability in regards the everyday German language (CH-GER).

The difficulty I see with table 2 is lack of information in regards exposure to CH-GER by the French speakers – the table seems to quantify exposure to stGER (at least that what the text in manuscript seems to imply). Would not French speakers have exposure to CH-GER through TV/radio, even if small quantity?

Author Response

Reviewer 1

# 1.The issue of gender distribution is confusing (page 7) – at one point it is stated there is difference in gender distribution between both groups and at other point it is stated there is not. Please include gender ratio in table 2. The statistical test needs to be corrected – a t-test was used to compare gender rations; in reality a chi-test needs to be performed.

We thank the reviewer for this remark, there was an initial confusion, we corrected it. We in fact did a chi square-test what showed no statistical difference in gender, but this was not reported correctly. We added the data to the table and corrected the statement to that there was no differences. Now it should be correct and less confusion. The gender matter is now in the table 2 and in the text, and in the legend of table2 it is explained.

“Statistics are calculated with dependent t-test, except for gender (chi square. *in gender the numbers stand for the number of females and males per group.)”

# 2. Do the French speakers learn stGER or CH-GER? If they only learn stGER they may still be familiar with CH-GER because they are living in a bilingual environment and thus have exposure to CH-GER and probably acquired knowledge). Did the authors acquire data in regards CH-GER ability (or comprehension ability) in the French speakers? The authors conclude that lack of differences means that diglossic ability does not seem to alter grey matter. But it may be that the French speakers do have some ability in regards the everyday German language (CH-GER).

This issue is effectively important, and we thank the reviewer to have raised it. French speakers learn only stGER in school. In the city of Fribourg, the predominantly spoken language is French and the exposure to CH-GER is very little. All the lectures, seminar, and clinical laboratories are done in French or in standard German, so university students communicate in French, standard German, or English. Moreover, our question was particularly focused on early diglossic, which was not the case of the French speaking group and if so, it was an exclusion factor.  Immersion in Swiss German occur during professional life. We did not specific asked for exposure of French CH to CH German since we were interested in the early exposure to diglossia.

 But we agree that there might be passive contact with CH-GER in the French group. As further studies we noted that we would have to do similar analysis comparing German speaking people living in Germany with CH-GER speaking people from Switzerland to exclude the influence of the exposure to the varieties of the Germans in the French speaking group.

To integrate this point, we have_

  • We specified “early diglossic speakers “in the title and in the abstract, as well as in the introduction since the question concerns participants who are early diglossics
  •  
  • We added in the introduction page 3-4: Their immersion CH-FR participants in CH-GER is negligible during studies in Fribourg and begins only later in life as in professional life. In the area of Fribourg, the language spoken outside university is predominantly French. In the CH-FR there are no differences in the spoken and the literary French. So, between these bilingual / multilingual groups the major difference is the early diglossia which is present only in the CH-GER group.
  • We added on the last paragraphe of the introduction, page 4: The question addressed here is whether early diglossia, in a bilingual context (French and German), leads to modification in brain structure.

# 3. The difficulty I see with table 2 is lack of information in regards exposure to CH-GER by the French speakers – the table seems to quantify exposure to stGER (at least that what the text in manuscript seems to imply). Would not French speakers have exposure to CH-GER through TV/radio, even if small quantity?

In the questionary we asked for the exposure of German. We agree that there is a lack of clear definition. Because the French-speaking participants that we included are exposed to CH-GER only later in life we predict that the exposure of CH-GER is very. There is no written material in CH-GER, no topic at university course and seminar in Swiss German nor is there a significant national exposure threw media (information is in St Ger).  (Se also point 2)

We have coined this point by inserting in the methods.

  • The… second group CH-FR native speaking with no exposure to CH-GER at school and negligible exposure during university (French participant living with CH-GER students were not included)
  • In Table 2 a line “CH-German” as spoken language, to confirm that No CHFR- spoke CH-GER as first or second language.
  • At the End of the introduction, we added: Their immersion CH-FR participants in CH-GER is negligible during studies in Fribourg and begins only later in life as in professional life. In the area of Fribourg, the language spoken outside university is predominantly French. In the CH-FR there are no differences in the spoken and the literary French. So, between these bilingual / multilingual groups the major difference is the early diglossia which is present only in the CH-GER group.

Reviewer 2 Report

Comments and Suggestions for Authors

Overall, while the study contributes to understanding the neurobiological underpinnings of language variation, it falls short of providing definitive conclusions about the impact of diglossia on brain structure. Here are the concerns:

1. The brain exhibits nuanced differences in its processing of different languages. Therefore, for a more effective exploration of diglossia, it is advisable to compare individuals proficient in German, both diglossic and non-diglossic, to elucidate potential neurocognitive distinctions.

2. Another issue present in the study is that all participants are bilingual. If the structural changes in the brain due to diglossia align with those observed in bilingualism, the results may potentially indicate no distinction in brain structure between diglossic individuals and non-diglossic individuals . The actual question addressed by the authors is whether diglossia, under bilingual conditions, leads to alterations in brain structure.

3.The determination of sample size necessitates a comprehensive consideration of multiple factors, including study design, anticipated effect size, statistical significance level, statistical power, and expected variability. Solely relying on the references provided by the authors, I cannot conclusively assert that a sample size of 24 participants is sufficient. I kindly request the authors to provide further clarification and explanation regarding this matter.

4.One potential explanation for the absence of differences found by the authors could be the inadequacy of the sample size relative to the effect size of diglossia.

5. In this paper, the authors selected only a few brain regions for comparison. However, conducting comparisons from the perspective of voxel or cortical surface vertex may yield different results.

Author Response

# 1. The brain exhibits nuanced differences in its processing of different languages. Therefore, for a more effective exploration of diglossia, it is advisable to compare individuals proficient in German, both diglossic and non-diglossic, to elucidate potential neurocognitive distinctions.

Thank you for this interesting comment. We agree that this is a factor that might influence the results, and we reviewed the literature. One study of Saur et al suggest that early French German bilinguals have similar brain activation in both language task on the contrary of late bilinguals. They do however show some difference in brain activation during specific syntactic judgement tasks (Saur D, Baumgaertner A, Moehring A et al. Word order processing in the bilingual brain. Neuropsychologia. 2009 47(1):158-68 – we can add it in the references if you think it is important). So, we agree that in further analysis we would need to compare the diglossic CH-GER speaking population with only German speaking population, as it is for people living in Germany. Unfortunately, this goes beyond the scope of this work and our available resources. Moreover, the schooling programs may present differences between countries. This is why we propose to use this paradigm for future studies and added this thought to ideas for research in the future in the conclusion part.

“…. and a            comparison of participants with similar standard L1, such as diglossic CH-/stGER and non-diglossic stGER speakers, will have to take place.

# 2. Another issue present in the study is that all participants are bilingual. If the structural changes in the brain due to diglossia align with those observed in bilingualism, the results may potentially indicate no distinction in brain structure between diglossic individuals and non-diglossic individuals. The actual question addressed by the authors is whether diglossia, under bilingual conditions, leads to alterations in brain structure.

Thank you for this statement. We agree that we did our study based on bilinguals / multilingual participants what might be a limitation. We tried to precise our question. Nevertheless, the two groups did not differ in the number of spoken languages, excluded CH-GER, and we controlled the age of acquisition, and immersion of the two groups. All participants were late bilinguals and the only difference between the two groups was early diglossia in the CH-GER. In theory this would be very interesting to perform an analysis comparing monolingual CF Fr, but we also must acknowledge that this is quite hard to find monolinguals in Switzerland in a university context because nowadays almost every person knows at least little of a second language.

We added then a sentence at the end of the introduction, page 4: The question addressed here is whether early diglossia, in a bilingual context (French and German), leads to modification in brain structure.

# 3.The determination of sample size necessitates a comprehensive consideration of multiple factors, including study design, anticipated effect size, statistical significance level, statistical power, and expected variability. Solely relying on the references provided by the authors, I cannot conclusively assert that a sample size of 24 participants is sufficient. I kindly request the authors to provide further clarification and explanation regarding this matter.

We thank the reviewer for the remark. We used the Cohen’s d effect size as a value for the G*power from an analysis of Mechelli et al., who is one of the seminal papers on brain structural differences between monolingual and bilinguals. We added this information into the methods and completed our analysis in the methods with our G power analysis (We used one tail comparisons, since MRI structural comparisons are one Tail).

Methods

Concerning the power analysis, we first referred to one of the seminal papers on brain structural differences between monolingual and bilingual, the Mechelli’s et al. analysis [21]. In their research, testing for differences in the density of grey and white matter between bilinguals and monolinguals, they recruited 25 monolinguals who have had little or no exposure to a second language and 25 early bilinguals; they found specific differences in one predicted area (increased left inferior parietal cortical density in bilinguals). We then made an analysis of statistical power with G Power paradigm [52]. To detect differences between two groups concluded, that, to achieve a power of 0.80 (One tail, effect Size d= 0.80, α = 0.05) a sample size of 21 participants per group should be used. So, we concluded that our sample was adequate compared to the bilingual literature.

# 4.One potential explanation for the absence of differences found by the authors could be the inadequacy of the sample size relative to the effect size of diglossia.

We agree with the reviewer. With looked at the comparison of Munson et al. [82]. According to their calculation of the Cohend’s d effect size to detect any difference, to be able to detect a medium effect size we would have to have a total group size of 218. So, to sum up: with our sample size (44 participants) we conclude that there is no large difference between our two groups. In order to detect minimal difference, we would need a much larger participants data set. We added this discussion and limitation to the paper (page 11).

There might be a lack of power due to small simple size. With the comparison of Munson et al. [82], mentioned before, and the calculation of the Cohen’s d effect size to detect any difference, we would have to have a total group size of 218 to detect a medium effect size. So, to sum up: with our sample size (44 participants) we conclude that there is no large difference between our two groups. If we want to detect any small difference between the two groups a five times larger sample size would be needed. We acknowledge, that the study of Munson et al. [82] is an analysis based on bilinguals and monolinguals and we have a diglossic dataset. Nevertheless, it is close to our question. If there was an eloquent difference between our two groups, we would be able to detect a difference based on the number of participants. The fact that Mechelli could detect differences in bilinguals with a number of participants similar to ours suggests, we believe that our sample size may be adequate to detect significant differences. Indeed, to detect minimal difference, we would need a much larger participants dataset.

# 5. In this paper, the authors selected only a few brain regions for comparison. However, conducting comparisons from the perspective of voxel or cortical surface vertex may yield different results.

 We were probably not clear enough in our methods parts. In fact, we did two different analyses, one on the whole brain and one on a group of regions of interests according essentially to Green and Abutalebi theories. We were conscious that a voxel-wise or a mean-wise analysis could lead to different results. It is why we have also conducted a voxel-wise analysis restricted in our region of interest in addition to the whole brain analysis. The analysis did not show any significant result either for the whole brain analysis or the regions of interest with a statistical threshold of pFWE< 0.05 family-wise error corrected for multiple comparison at a peak level with the Threshold-Free Cluster Enhancement Estimator (TFCE toolbox under SPM12) as for the whole brain. We have added a small introductory paragraph in the Analyses of the Methods

We conducted two different structural analyses: Voxel-Based Morphometry (VBM) and Surface-Based Morphometry (SBM). Each of them was conducted on the whole brain and on a group of regions of interests according essentially to Green and Abutalebi theories. Moreover, a voxel-wise or a mean-wise analysis could lead to different results. That is why we decided to also conduct a voxel-wise analysis restricted in our region of interest in addition to the whole brain analysis.

Reviewer 3 Report

Comments and Suggestions for Authors

The present study compared the cortical strutures between Diglossic and non-diglossic groups. The results showed no differences between the two groups, providing the initial evidence of the influence of diglossia on brain structures. 

The present study has some contribution to our understanding of bilingualism and diglossia. Some issues should be noted: 

1. The authors should have more theoretical contribution to bilingual and cognitive advantages, such as IC model (Green, 1998) and bilingual advantage debate. How can the results from the present study could inform these theories or debates? 

2. Why did the authors test the cognitive abilities of two groups, such as inhibition and task switching abilities. I suggest the authors at least mention this as a limitation, because the present study only have the cortical information but seledom connect with the cognitive or linguistic performance. 

3. Based on the present study, how can the authors discuss the representation of bilingualism and diglossia? Any differences or similarities? Can models (e.g., RHM, BIA) in bilingualism be implied to diglossia? Any alternations or changes? 

Comments on the Quality of English Language

The English is fine. 

Author Response

# 1. The authors should have more theoretical contribution to bilingual and cognitive advantages, such as IC model (Green, 1998) and bilingual advantage debate. How can the results from the present study could inform these theories or debates? 

Thank you for this interesting point, we added some more theoretical inputs in the introduction explaining the model of IC as well as the impact on cognitive control systems.

Communicating in a bilingual context needs to control which language must be used and which not. According to the Inhibition Control (IC) model of Green, 1998) [83], a bilingual must inhibit the non-target language in order to communicate in the other one. This mechanism also implies the ability to switch between languages and activating the language needed [49]. The process of switching seems to share features with other cognitive control systems [12].

#2. Why did the authors test the cognitive abilities of two groups, such as inhibition and task switching abilities. I suggest the authors at least mention this as a limitation, because the present study only have the cortical information but seldom connect with the cognitive or linguistic performance. 

Effectively, did not look at cognitive abilities as in task switching and inhibition, that is correct. This would be interesting, but we focused on the structural changes, so we did not add this question. We added this statement to the discussion -limitation.

Finally, we did not compare executive abilities between both groups. This could have given us supplementary measures of an eventual impact of diglossia on brain function.   Such cognitive approach would be of high interest in future studies, given the established relationship between language control and executive function.

# 3. Based on the present study, how can the authors discuss the representation of bilingualism and diglossia? Any differences or similarities? Can models (e.g., RHM, BIA) in bilingualism be implied to diglossia? Any alternations or changes? 

This effectively an important point, which is not directly tested in our study, but which can certainly be integrated in the discussion.  Actually, in the IC model, the dominance of one language influences This the switching cost. When both languages become identical as in the difference between the dominance becomes small, they have a comparable switching cost and more automatized control system. This has been studied in balanced bilinguals. Adjusting this to the situation of diglossia, the two oral languages are very similar and can be considered linked in a L1 entity. Moreover, there is only one written language. Thus, the active inhibition processes might be considered minimal and lead to no structural differences. Referring to the BIA model, baseline activation could be identical for target and nontarget L1 lexicon of CH-GER and stGER.. We References for these models to the introduction as well as we used it for our discussion of our results.

Introduction, page 1-2

Communicating in a bilingual context needs to control which language must be used and which not. According to the Inhibition Control (IC) model of Green, 1998) [83], a bilingual must inhibit the non-tagret language in order to communicate in the other one. This mechanism also implies the ability to switch between languages and activating the language needed [49]. The process of switching seems to share features with other cognitive control systems [12]. Another model, the Bilingual Interactive Activation (BIA), has been developed in language comprehension, and proposes that words of both languages are stored in a single, shared lexicon, and that similar words in both languages, the relevant and the irrelevant one, get activated when a word is presented [96, 97]. Growing literature suggests that bilingualism can be considered as a spectrum with multiple components, such as age of second language acquisition (L2 AoA), L2 exposure, and proficiency [4-6]. The proficiency also influences the amount of inhibitory control that is activated, as explained in the IC-model [83]. With an increased proficiency in L2 there is a shift to a more automatic control system and easier language processing [84].

Discussion, page 9-10

In the IC model of Green (1998) it has been predicted that depending on the dominance of the language, inhibition takes place as there is a larger switching cost for the stronger language [91-93]. On the other hand, it has been said that when the dominance between L1 and L2 is similar, the switching costs are comparable [94, 95]. We suggest that, in diglossia, both CH-GER and stGER can be considered L1 depending on the situation either one is needed, this then would lead to minimal switching costs, so minimal structural changes in the associated brain areas. Also, Diglossia may also apply to the BIA model [96, 97], where a similar baseline activation can occur and lexical differences are minimal between both L1 (CH-GER and stGER). Therefore, the word activation is determined by the context. The strong link between CH-GER and stGER can also justify why there is little mediation mechanism, in the sense proposed by Revised Hierarchical Model (RHM) [100]. Also, the conceptual system of the two diglossic lexicons is highly linked, and producing a word in stGER does not need to be mediated by CH-GER, like it would be the case in less close L2 [100, 101]. The question of intralanguage control mechanisms must be further studied.

Round 2

Reviewer 2 Report

Comments and Suggestions for Authors

The issues identified in the submitted article have been adequately addressed.

Author Response

Thank you.

Reviewer 3 Report

Comments and Suggestions for Authors

I think the authors addressed my concerns. 

Comments on the Quality of English Language

Englis is fine. I suggest the authors finally review and proofread the MS before acceptance. 

Author Response

Thank you.